# Core Promoter and Pre-Core Variants of the Hepatitis B Virus (HBV) Are Frequent in Chronic Hepatitis B HBeAg-Negative Patients Infected by Genotypes A and D

**DOI:** 10.3390/v15122339

**Published:** 2023-11-29

**Authors:** Tania Reuter, Michele Soares Gomes-Gouvea, Samira Chuffi, Ulisses Horst Duque, Waltesia Perini, Raymundo Soares Azevedo, João Renato Rebello Pinho

**Affiliations:** 1Internal Medicine Department, Health Science Center, University Hospital Cassiano Antônio de Moraes, Federal University of Espirito Santo, Vitória 29041-295, ES, Brazil; ulissesduque@terra.com.br (U.H.D.); waltesia68@hotmail.com (W.P.); 2LIM-07, Institute of Tropical Medicine, Department of Gastroenterology, School of Medicine, University of Sao Paulo, São Paulo 05403-907, SP, Brazil; gomesmic@yahoo.com.br (M.S.G.-G.); sa.chuffi@gmail.com (S.C.); joao.pinho@einstein.br (J.R.R.P.); 3Department of Pathology, School of Medicine, University of Sao Paulo, São Paulo 01246-903, SP, Brazil; razevedo@usp.br; 4Clinical Laboratory, Hospital Israelita Albert Einstein, São Paulo 05652-900, SP, Brazil; 5LIM-03, Central Laboratories Division, Clinics Hospital, School of Medicine, University of Sao Paulo, São Paulo 05403-000, SP, Brazil

**Keywords:** chronic hepatitis B, HBV, pre core mutations, basal core promoter mutations, BCP and PC variants

## Abstract

In Brazil, hepatitis B virus endemicity is low, moderate, or high in some areas, such as Espírito Santo State in the southeast region. In this study, we intend to characterize the basal core promoter (BCP) and pre-core region (PC) variants and their association with clinical/epidemiological disease patterns in patients infected with genotypes A and D. The study included 116 chronic hepatitis B patients from Espírito Santo State, Southeast Brazil, infected with genotypes A and D. Basal core promoter (BCP) and pre-core mutations were analyzed in these patients. The frequency of BCP and PC mutations was compared with age, HBeAg status, HBV genotype and subgenotype, HBV-DNA level, clinical classification, and transmission route. HBeAg-negative status was found in 101 (87.1%) patients: 87 (75.0%) were infected with genotype A (A1 = 85; A2 = 2) and 29 (25.0%) were infected with genotype D (D3 = 24; D4 = 3; D2 = 2). BCP + PC variants altogether were more frequent (48.1%) in genotype D than in genotype A strains (6.0%) (*p* < 0.001). When this evaluation was performed considering the cases that presented only the A1762T and/or G1764A (BCP) mutations, it was observed that the frequency was higher in genotype A (67.5%) compared to genotype D (7.4%) (*p* < 0.001). On the other hand, considering the samples with mutations only in positions G1896A and/or G1899A (PC), the frequency was higher in genotype D (75.8%) than in genotype A (6.9%) (*p* < 0.001). Interestingly, HBV DNA was lower than 2000 IU/mL especially when both BCP/PC mutations were present (*p* < 0.001) or when only PC mutations were detected (*p* = 0.047), reinforcing their role in viral replication.

## 1. Introduction

During long-term infection with hepatitis B virus (HBV), different selective pressures, particularly the immune system, shape viral populations through high rates of nucleotide substitutions and low revision capacity, enabling the emergence of variants with mutations in this pool of quasi-species [1,2].

Genotypes A, D, and F are the most prevalent in Brazil, reflecting the origin of the Brazilian population, descended from enslaved Africans, European colonizers, and originary people [3,4]. Mutations in the pre core (PC) region, particularly at position 1896 (G/A), generate early stop codons, whereas the classic mutations in the basal core promoter (BCP) region (i.e., double mutations at positions 1762 and 1764, the so-called TA change) are primarily substitutions [5]. These viral factors are associated with the outcome of HBV infection, including HBV genotypes, HBV DNA levels, clinical presentation, and the natural history of HBV infection, as it has been demonstrated that viral mutations in the BCP and/or PC regions affect replication and may increase liver injury [6]. In addition to these variants, other mutations, such as T1753V in the BCP region and G1899A and G1862T in the PC region, have been identified, which are associated with the development of HCC, HBeAg production, and the reverse transcription process, respectively [7,8]. The aim of this study was to investigate the genetic characteristics in the BCP and PC regions of the different HBV genotypes/subgenotypes found at Espírito Santo state, southeast Brazil, and to evaluate the association between these variants and demographic, epidemiological, clinical, and virological patterns.

## 2. Materials and Methods

### 2.1. Type of Study and Sample Population

The study design is a series of cases with retrospective evaluation of individuals enrolled in a hepatitis outpatient clinic at the University Hospital Cassiano Antonio de Moraes (HUCAM) from the Federal University of Espirito Santo, Vitória, Espírito Santo (ES) State, Brazil from July 2005 to July 2017. Demographic, epidemiological, clinical, treatment, and laboratory characteristics were detailed in a previous publication involving this same population [4]. Patients harboring HBV genotypes A and D and their subgenotypes were characterized by amplification and sequencing of the S region and polymerase of the HBV genome (S/POL), as described by Gomes-Gouvêa et al., 2015 [9]. In the present study, we analyzed the BCP and PC regions from HBV isolated from these patients. For statistical analysis, the clinical phase of hepatitis B considered in this study was the one described in the medical record in July 2017.

### 2.2. Characterization of Variants in the BCP and PC Regions

For the amplification of the BCP and PC regions of the HBV genome, nested PCR methodology was applied. Primers 2032R and EP 1.1 were used for the first amplification step, generating a fragment of 554 bp, and primers 2017R and EP 2.1 were used for the second amplification step, generating a fragment of 501 bp [10]. The sequence was characterized by Sanger methodology using ABI Prism^®^ BigDyeTM Terminator Cycle Sequencing Ready Reaction Kit (Applied Biosystems, Foster City, CA, USA). The search for mutations in the BCP and PC regions was performed by visual analysis of the alignment of nucleotide sequences. In the BCP and PC regions, we focused on nucleotide positions 1762 and 1764, 1814–1816, 1858, 1862, 1888, 1896, and 1899. The following substitutions were considered: A1762T and G1764A; any change that promotes the alteration of the ATG initiation codon in positions 1814–1816; in position 1858, any nucleotide found; G1862T; G1888A; G1896A; and G1899A. Further analyses were performed for BCP (A1762T; G1764A) and PC (G1896A; G1899A) mutations, which are classically associated with reduced or absent HBeAg synthesis in genotype A and genotype D, respectively.

### 2.3. Statistical Analysis

The frequency of BCP and PC mutations was compared to age, HBeAg status, HBV genotype and subgenotype, HBV-DNA level, clinical classification, and transmission route. The significance level used in all statistical tests was 5% (α = 0.05).

The results were analyzed using the statistical programs GraphPad Prism (Software Inc., San Diego, CA, USA, version 5.01), Minitab version 17.3.1 (Minitab Inc., State College, PA, USA), and SPSS (version 23.0, IBM SPSS^®^ Statistics, Chicago, IL, USA). When indicated, the following analyses of the results were carried out: mean, median, standard deviation, Fisher’s exact test, Chi-square test, Mann–Whitney non-parametric test, and Kruskal–Wallis non-parametric test [11]. 

## 3. Results

In this study, HBVDNA-positive samples from chronic hepatitis B patients were subjected to amplification and Sanger sequencing of the BCP and PC regions. Among these samples, it was possible to amplify and sequence the BCP and PC regions to search for mutations in samples from 111 and 116 patients, respectively. Among the 116 chronic HBV carriers that were analyzed in this study, 87 (75.0%) were infected with genotype A (A1 = 85; A2 = 2) and 29 (25.0%) were infected with genotype D (D3 = 24; D4 = 3; D2 = 2). Among these cases, individuals >40 years (62.9%—73/116) predominated, with probable intrafamilial transmission in 60.3% (70/116). HBeAg status was negative in 87.1% (101/116) and positive in 12.9% (15). Regarding the stages of chronic hepatitis B, 42.2% (49/116) of individuals had active disease. 

Table 1 shows the frequency of mutations in the BCP region among the 111 samples analyzed. Variants A1762T and/or G1764A were observed in 68.4% of the samples. The G1764A mutation was observed more frequently (68.2%) than the A1762T (54.0%) and 52.7% of the samples had the A1762T + G1764A association. Our study showed that the frequency of these mutations was always lower in genotype D, but only the lower frequency of the G1764A mutation (51.8%) was statistically significant (*p* = 0.04). This significance was confirmed (*p* = 0.034) when we analyzed the subgenotypes (A1, D3, and others (A2, D2, and D4)). In the ES population, as shown in Table 1, none of these mutations seem to play any role in relation to HBeAg status.

In the analysis of PC region variability, 24.1% of the samples had mutations at positions 1896 and/or 1899 (Table 2). The G1899A mutation was slightly more frequent (17.2%) than the G1896A (16.4%), and 9.5% of the samples had both mutations (G1896A/G1899A). Mutations in the PC region were more frequent in genotype D (75.8%) than in genotype A (6.9%) and this difference was statistically significant (*p* < 0.001). This significant difference persisted when assessing these mutations individually or in combination, and it was validated in the comparison between subgenotypes A1 and D3 (*p* < 0.001).

The presence of strains with mutations in the PC region showed no association with the age of the patients, transmission route, clinical form of hepatitis B, or HBeAg status. On the other hand, intermediate values of HBV DNA (2000 IU/mL–10,000 IU/mL) showed statistical significance with the presence of the isolated G1896A mutation (*p* = 0.046) or in association with the G1899A mutation (*p* = 0.014).

Table 3 shows the analysis of mutations in the BCP + PC regions altogether. These variants were more frequent (48.1%) in genotype D than in genotype A strains (6.0%) (*p* < 0.001). When this evaluation was performed considering the cases that presented only the A1762T and/or G1764A (BCP) mutations, it was observed that the frequency was higher in genotype A (67.5%) compared to genotype D (7.4%) (*p* < 0.001). When this evaluation was performed considering the samples with mutations only in positions G1896A and G1899A (PC), the frequency was higher in genotype D (29.6%) than in genotype A (1.2%), with statistical significance (*p* < 0.001). Interestingly, HBV DNA was lower than 2000 IU/mL especially when both BCP/PC mutations were present (*p* < 0.001) or when only PC mutations were detected (*p* = 0.047).

The presence of clinically relevant mutations in other positions of the PC gene, such as 1814–1816, related to the HBeAg synthesis initiation codon, in addition to 1862 and 1868, were also evaluated and are shown in Table 4. 

The frequency of samples with mutations in 1814–1816 was 19.0% (22/116), in G1862T, it was 71.5% (83/116), and in G1888A, the frequency of C or T mutations was 43.1% (50/116). This latter one was associated with age between 20–40 years (*p* = 0.034). 

These PC region mutations were more frequent in genotype A than in genotype D, which was confirmed by the evaluation of subgenotypes A1 and D3 (Table 4). The G1862T and G1888A/C/T mutations were more frequent in genotype A (or subgenotype A1) than in genotype D (or subgenotype D3) with statistical significance (*p* < 0.001). Likewise, the presence of strains with mutations at positions 1814–1816, 1862, and 1888 of the PC region was associated with negative HBeAg clinical forms (*p* = 0.03), with them being more frequent in inactive carriers (26.7%) followed by individuals with active HBeAg-negative chronic hepatitis (14.6%).

## 4. Discussion

Classic variants of the BCP and PC regions have been described during the course of chronic HBV infection. Among these mutations, those in the BCP (1762/1764) and PC (1896/1899) regions are generally associated with lower expression of the HBeAg antigen. Although HBeAg is an accessory protein and is not necessary for viral replication, it can moderate immune activity, facilitating chronicity and favoring the persistence of infection in the individual. Strains with mutations in the BCP and/or PC regions, strongly described in genotypes B and C, are commonly responsible for the continuous replication of HBV after seroconversion of HBeAg and consequent maintenance of inflammatory activity in the liver [12]. However, evidence on other genotypes is still limited.

Gunther et al., 1998 [7] observed that these mutations reduced pre-C mRNA levels and HBeAg secretion by 55% compared to the wild-type virus. On the other hand, Jammeh et al., 2008 [13] showed that the presence of the mutation A1762T and G1764A, isolated or associated, replicated with an efficiency equivalent to that of the wild virus but that the presence of other mutations such as C1766T, T1768A, and T1753C leads to a virus with a relevant reduction in its replication efficiency. The author concludes that mutations in the BCP region other than those in 1762/1764 appear to upregulate viral replication and, at the same time, significantly reduce HBeAg production. These results suggest that the genetic properties of HBV and the interactions of cellular proteins with the region of mutations may play an important role in the pathogenesis of hepatitis B [14]. Interestingly, among our samples, HBV DNA was lower than 2000 IU/mL especially when both BCP/PC mutations were present (*p* < 0.001) or when only PC mutations were detected (*p* = 0.047), reinforcing their role in viral replication.

In Espírito Santo, BCP mutations A1762T/G1764A were more frequent in genotype A than D while G1896A/G1899A mutations were more frequent in genotype D. The presence of the mutations 1814–1816, G1862T, and G1888A/C/T did not have any influence on the HBV DNA levels. Curiously, G1862T and G1888A/C/T were more frequent in genotype A (or subgenotype A1) samples. The most striking point was that 1814–1816 mutations were not found in HBeAg-positive samples, which is expected from mutations that alter the start codons for the synthesis of HBeAg.

In our study, we found a higher frequency of the G1764A mutation associated with genotype A (*p* = 0.040) or subgenotype A1 (*p* = 0.034). We did not find similar results in the literature. We cannot explain why this mutation pattern was so frequent in our study. It is possible that there are different binding proteins in HBV carriers that can influence the establishment of this mutation. Further studies, especially with next-generation sequencing in this population, could help to clarify this point. 

We have found many cases where only the G1764A mutation was present without the A1762T mutation. The discovery of this isolated mutation is important as it is rarely described. It would be interesting to look for this pattern in other populations to see if we can better discover the origin of the circulating HBV in Espírito Santo.

The A1762T/G1764A mutations together are associated with an important reduction in pre-C mRNA levels and HBeAg synthesis [7]. The isolated G1764A mutant was also the focus of one study, and it was demonstrated that constructs bearing an A1762T or G1764A substitution, or the two together, replicated with an efficiency equal to that of the WT. On the other hand, three constructs, with the substitutions C1766T/T1768A, A1762T/G1764A/C1766T, and T1753C/A1762T/G1764A, released more virions, whilst the T1753C/A1762T/G1764A/C1766T construct released significantly fewer. Thus, core promoter mutations other than those at 1762/1764 appear to upregulate viral DNA replication and, at the same time, reduce HBeAg production [13]. These mutations play a role in the interaction of cellular proteins with HBV, such as HNF4 [14,15]. These mutations associated with others present in other regions play an important role in HBV replication [16]. These mutations were found in the studied population, but we did not find any correlation among them with age, HBeAg status, HBV genotype and subgenotype, HBV-DNA level, clinical classification, and transmission route.

The A1762T/G1764A mutations in the BCP region show great variability among the genotypes. Ghosh et al., 2012 [17] found this double mutation in 27.5% of genotype D individuals. Similarly, a recent study in a population with various ethnic groups found this double mutation in 24.2% of the sample [18]. On the other hand, a recent Canadian study found much higher frequencies (62.5%) than a previous study in that country, attributing this increase to strong immigration, especially from Asia, that introduced HBV genotypes with different distributions of BCP and PC mutations [5].

In Brazil, studies evaluating double mutations in the BCP region are scarce and present great variability, reporting lower frequencies (33.3%) in genotype B and higher frequencies in genotypes A and F (62% and 90%, respectively) [10]. 

Similarly to our results, genotype A was the most frequent in the Northeast region [19], while genotypes A and D were reported as the most frequent in the south and southeast regions [20,21].

In our study, the double mutation A1762T/G1764A was frequent (52.7%) and showed no significant difference between genotypes. In addition, an analysis was performed between the most frequent subgenotypes, A1 and D3, with clinical phases of chronic hepatitis B, and, again, no statistically significant difference was found. To date, this is the first study to evaluate in more detail the frequency of these mutations in subgenotypes in Brazil.

Few studies have evaluated the behavior of PC region variants circulating in Brazil and their influence on liver disease in patients with chronic hepatitis B. Rezende et al. (2005) [22], identified that the presence of the G1896A mutation was associated with more severe liver injury. On the other hand, Chacha et al. (2017) [21] identified a frequency of this variant associated with genotype D, but not with more severe liver disease.

In our study, no significant difference was found between the presence of the G1896A mutation and epidemiological characteristics (age and transmission route), HBeAg status, or clinical phase of HBV infection. These data support the hypothesis of a mix of strains with and without these mutations in the circulating virus pool during the natural course of chronic hepatitis B [23]. It would be important to look for this mutation pattern in other regions of the world, particularly in Italy, Portugal, and Africa, from where most of the population of ES came. 

The clinical significance of the presence of non-classical mutations in the PC region, such as G1862T and G1888A/C/T, has been poorly characterized. In our study, an association was found between the PC G1888A/C/T variant and an age of less than 40 years (52.7%). One hypothesis for this finding could be the occurrence of this mutation at an early age, lower expression of the HBeAg antigen, and escape from the host’s immune system with incomplete viral clearance, thereby maintaining chronic infection with HBV. The presence of the second variant, G1862T, was associated with genotype A, confirmed in subgenotype A1. This association has been previously described by Kramvis et al. (2013) [24] in South Africa and recently by Chacha et al. (2017) [21] in Brazil. However, its value in the natural history of chronic hepatitis B is still unknown.

## 5. Conclusions

BCP mutations A1762T/G1764A were more frequent in genotype A than D while G1896A/G1899A mutations were more frequent in genotype D. Interestingly, HBV DNA was lower than 2000 IU/mL especially when both BCP/PC mutations were present (*p* < 0.001) or when only PC mutations were detected (*p* = 0.047), emphasizing their role in viral replication. The presence of the mutations 1814–1816, G1862T, and G1888A/C/T had no effect on HBV DNA levels. Curiously, the non-classical PC mutations, G1862T and G1888A/C/T, were more frequent in the genotype A (or subgenotype A1) samples.

## Figures and Tables

**Table 1 viruses-15-02339-t001:** Frequency of mutations in the BCP region of HBV genomes isolated from individuals with chronic hepatitis B from Espírito Santo State, Brazil, and its association with demographic, clinical, and other virological characteristics.

Variable	Total	A1762T *n* = 60	*p*-Value A1762T	G1764A *n* = 75	*p*-Value G1764A	A1762T + G1764A *n* = 58	*p*-Value A1762T + G1764A	A1762T and/or G1764A *n* = 76	*p*-Value A1762T and/or G1764A	WT BCP *n* = 34	*p*-Value WT BCP
	*n* = 111	54.0%		68.2%		52.7%		68.4%		30.6%	
Age (years)											
20–40	41 (36.9%)	22 (53.7%)	0.949	27 (67.5%)	0.908	21 (52.5%)	0.971	27 (67.5%)	0.785	13 (32.5%)	0.785
>40	70 (63.1%)	38 (54.3%)		48 (68.6%)		37 (52.8%)		49 (70%)		21 (30%)	
HBeAg status											
Negative	97 (87.4%)	51 (52.3%)	0.407	66 (68%)	0.931	50 (51.5%)	0.496	67 (69%)	0.991	30 (30.9%)	0.991
Positive	14 (12.6%)	9 (64.3%)		9 (69.2%)		8 (61.5%)		9 (69.2%)		4 (30.7%)	
Genotype											
A	84 (75.7%)	48 (57%)	0.250	61 (73.5)	0.040	47 (56.6%)	0.151	61 (73.5%)	0.086	22 (26.5%)	0.086
D	27 (24.3%)	12 (44.4%)		14 (51.8%)		11 (40.7)		15 (55.6%)		12 (44.4%)	
Subgenotype											
A1	82 (73.9%)	47 (57.3%)	0.497	60 (74%)	0.034	46 (56.8%)	0.360	60 (74%)	0.080	21 (25.9%)	0.080
D3	24 (21.6%)	11 (45.8%)		11 (45.8)		10 (41.7%)		12 (50%)		12 (50%)	
Others (A2, D2, D4)	5 (4.5%)	2 (40%)		4 (80%)		2 (40%)		4 (80%)		1 (20%)	
HBV DNA (IU/mL) *											
Up to 2000	47 (42.3%)	25 (53.2%)	0.981	32 (68.1%)	0.398	24 (51%)	0.933	33 (70.2%)	0.395	14 (29.8%)	0.395
From 2001 to 10,000	27 (24.3%)	15 (55.6%)		21 (77.8%)		15 (55.6%)		21 (77.8%)		6 (22.2%)	
Higher than 10,000	35 (31.5%)	19 (54.3%)		21 (61.8%)		18 (52.9%)		21 (61.8%)		13 (38.2%)	
Clinical classification											
HBeAg-positive chronic infection	7 (6.3%)	4 (57.1%)	0.708	4 (57.1%)	0.671	4 (57.1%)	0.850	4 (57.1%)	0.617	3 (42.9%)	0.609
HBeAg-positive chronic hepatitis	7 (6.3%)	5 (71.4%)		5 (83.3%)		4 (66.7%)		5 (83.3%)		1 (16.7%)	
HBeAg-negative chronic hepatitis	39 (35%)	19 (48.7%)		25 (64%)		19 (48.7%)		25 (64.1%)		14 (35.9%)	
HBeAg-negative chronic infection	58 (52.3%)	32 (55.2%)		41 (70.7%)		31 (53.4%)		42 (72.4%)		16 (27.6%)	
Transmission route											
Intrafamilial	68 (61.3%)	36 (52.9%)	0.767	45 (67.2%)	0.775	34 (50.7%)	0.603	46 (68.7%)	0.902	21 (31.3%)	0.902
Others	43 (38.7%)	24 (55.8%)		30 (69.7%)		24 (55.8%)		30 (69.8%)		13 (30.2%)	

* missing viral load data in two patients.

**Table 2 viruses-15-02339-t002:** Frequency of mutations in the PC region of HBV genomes isolated from individuals with chronic hepatitis B from Espírito Santo State, Brazil, and its association with demographic, clinical, and other virological characteristics.

Variable	Total	G1896A *n* = 19	*p*-Value G1896A	G1899A *n* = 20	*p*-Value G1899A	G1896A + G1899A *n* = 11	*p*-Value G1896A + G1899A	G1896A and/or G1899A *n* = 28	*p*-Value G1896A and/or G1899A	WT PC *n* = 88	*p*-Value WT PC
	*n* = 116	16.4%		17.2%		9.5%		24.1%		75.9%	
Age (years)	*n* = 116	19		20		11		28		88	
20–40	43 (37.1%)	5 (11.6%)	0.279	6 (13.9%)	0.466	3 (7%)	0.471	8 (18.6%)	0.279	35 (81.4%)	0.279
>40	73 (62.9%)	14 (19.2%)		14 (19.1%)		8 (11%)		20 (27.4%)		53 (72.6%)	
HBeAg status	*n* = 116										
Negative	101 (87.1%)	17 (16.8%)	1.000	19 (18.8%)	0.462	10 (9.9%)	1.000	26 (25.7%)	0.517	75 (74.3%)	0.517
Positive	15 (12.9%)	2 (13.3%)		1 (6.7%)		1 (6.7%)		2 (13.3%)		13 (86.7%)	
Genotype	*n =* 116										
A	87 (75%)	1 (1.1%)	<0.001	5 (5.7%)	<0.001	0	<0.001	6 (6.9%)	<0.001	81 (93.1%)	<0.001
D	29 (25%)	18 (62%)		15 (51.6%)		11 (37.9%)		22 (75.8%)		7 (24.1%)	
Subgenotype	*n* = 116										
A1	85 (73.3%)	1 (1.2%)	<0.001	5 (5.9%)	<0.001	0	Not determined	6 (7%)	<0.001	79 (92.9%)	<0.001
D3	24 (20.7%)	14 (58.3%)		15 (58.3%)		10 (41.7%)		18 (75%)		6 (25%)	
Others (A2, D2, D4)	7 (6%)	4 (57.1%)		1 (14.3%)		1 (14.3%)		4 (57.1%)		3 (42.9%)	
HBV DNA (IU/mL)	*n* = 114 *										
Up to 2000	50 (43.9%)	4 (8%)	0.046	6 (12%)	0.063	1 (2%)	0.014	9 (18%)	0.089	41 (82%)	0.089
From 2001 to 10,000	27 (23.7%)	8 (29.6%)		9 (33.3%)		6 (22.2%)		11 (40.7%)		16 (59.3%)	
Higher than 10,000	37 (32.4%)	7 (18.9%)		5 (13.5%)		4 (10.8%)		8 (21.6%)		29 (78.4%)	
Clinical classification	*n* = 116										
HBeAg-positive chronic infection	7 (6%)	0	0.243	0	0.377	0	Not determined	0	0.256	7 (100%)	0.256
HBeAg-positive chronic hepatitis	8 (6.9%)	2 (25%)		1 (12.5%)		1 (12.5%)		2 (25%)		6 (75%)	
HBeAg-negative chronic hepatitis	41 (35.3%)	9 (21.9%)		7 (17%)		5 (12.2%)		11 (26.8%)		30 (73.2%)	
HBeAg-negative chronic infection	60 (51.7%)	8 (13.3%)		12 (20%)		5 (8.3%)		15 (25%)		45 (75%)	
Transmission route	*n* = 116										
Intrafamilial	70 (60.3%)	14 (20%)	0.184	13 (18.6%)	0.638	8 (11.4%)	0.367	19 (27%)	0.346	51 (72.9%)	0.346
Others	46 (39.7%)	5 (10.9%)		7 (15.2%)		3 (6.5%)		9 (19.6%)		37 (80.4%)	

* missing viral load data in two patients.

**Table 3 viruses-15-02339-t003:** Frequency of mutations in the BCP + PC regions of HBV genomes isolated from individuals with chronic hepatitis B from Espírito Santo State, Brazil, and its association with demographic, clinical, and other virological characteristics.

Variable	Total	WT BCP+ WT PC *n* = 25	*p*-Value WT BCP+ WT PC	BCP Mutation (A1762T and/or G1764A) + PC (G1896A and/or G1899A) *n* = 18	*p*-Value BCP Mutation (A1762T and/or G1764A) + PC (G1896A and/or G1899A)	Only Mutation in BCP *n* = 58	*p*-Value Only Mutation in BCP	Only in PC *n* = 9	*p*-Value Only in PC
	*n* = 110	22.7%		16.4%		52.7%		8%	
Age (years)	*n* = 110	25		18		58		9	
20–40	40 (36.4%)	10 (25%)	0.669	5 (12.5%)	0.400	22 (55%)	0.718	3 (7.5%)	0.843
>40	70 (63.6%)	15 (21.4%)		13 (18.6%)		36 (51.4%)		9 (8.2%)	
HBeAg status	*n* = 110								
Negative	97 (88.2%)	21 (21.6%)	0.487	16 (16.5%)	1.000	51 (52.6%)	0.931	9 (9.3%)	0.595
Positive	13 (11.8%)	4 (30.8%)		2 (15.4%)		7 (53.8%)		0.000	
Genotype	*n* = 110								
A	83 (75.5%)	21 (25.3%)	0.242	5 (6.0%)	<0.001	56 (67.5%)	*<0.001*	1 (1.2%)	*<0.001*
D	27 (24.5%)	4 (14.8%)		13 (48.1%)		2 (7.4%)		8 (29.6%)	
Subgenotype	*n* = 110								
A1	81 (73.6%)	20 (24.7%)	0.692	5 (6.2%)	Not determined	55 (67.9%)	<0.001	1 (1.2%)	Not determined
D3	24 (21.8%)	4 (16.7%)		10 (41.7%)		2 (8.3%)		8 (33.3%)	
Others (A2,D2,D4)	5 (4.5%)	1 (20.0%)		3 (60.0%)		1 (20.0%)		0	
HBV DNA (IU/mL)	*n* = 108 *								
Up to 2000	47 (43.5%)	13 (27.6%)	0.010	7 (14.9%)	0.683	26 (55.3%)	0.722	1 (2.2%)	0.047
From 2001 to 10,000	27 (25.0%)	1 (3.7%)		6 (22.2%)		15 (55.6%)		5 (18.5%)	
Higher than 10,000	34 (31.5%)	10 (29.4%)		5 (14.7%)		16 (47.0%)		3 (8.8%)	
Clinical classification	*n* = 110								
HBeAg-positive chronic infection	7 (6.4%)	3 (42.9%)	0.533	0	Not determined	4 (57.1%)	0.927	0	Not determined
HBeAg-positive chronic hepatitis	6 (5.5%)	1 (16.7%)		0		3 (50.0%)		2 (33.3%)	
HBeAg-negative chronic hepatitis	39 (35.5%)	10 (25.6%)		4 (10.3%)		19 (48.7%)		6 (15.4%)	
HBeAg-negative chronic infection	58 (52.7%)	11 (19%)		5 (8.6%)		32 (55.2%)		10 (17.2%)	
Transmission route	*n* = 110								
Intrafamilial	67 (60.9%)	14 (20.9%)	0.569	11 (16.4%)	0.985	35 (52.2%)	0.898	7 (10.4%)	0.262
Others	43 (39.1%)	11 (25.6%)		7 (16.3%)		23 (53.5%)		2 (2.6%)	

* missing viral load data in two patient.

**Table 4 viruses-15-02339-t004:** Frequency of other clinically relevant mutations in the PC region of HBV genomes isolated from individuals with chronic hepatitis B from Espírito Santo State, Brazil, and its association with demographic, clinical, and other virological characteristics.

Variable	Total	1814–1816 (Changes in the Start Codon) *n* = 22	*p*-Value 1814–1816	G1862T *n* = 83	*p*-Value G1862T	G1888A, C, or T *n* = 50	*p*-Value G1888A, C or T
	*n* = 116	19.0%		71.5%		43.1%	
Age (years)	*n* = 116						
20–40	43 (37.1%)	9 (20.9%)	0.680	34 (79%)	0.162	24 (55.8%)	0.034
>40	73 (62.9%)	13 (17.8%)		49 (67%)		26 (35.6%)	
HBe Ag status	*n* = 116						
Negative	101 (87.1%)	22 (21.8%)	0.070	74 (73.3%)	0.358	43 (42.6%)	0.766
Positive	15 (12.9%)	0		9 (60.0%)		7 (46.7%)	
Genotype	*n* = 116						
A	87 (75.0%)	19 (21.4%)	0.150	82 (94.2%)	<0.001	50 (57.3%)	<0.001
D	29 (25.0%)	3 (10.3%)		1 (3.4%)		0	
Subgenotype	*n* = 116						
A1	85 (73.3%)	19 (22.3%)	0.118	82 (96.5%)	<0.001	50 (58.8%)	<0.001
D3	24 (20.7%)	3 (12.5%)		1 (4.2%)		0	
Others	7 (6%)	0		0		0	
HBV DNA (IU/mL)	*n* = 114 *						
Up to 2000	50 (43.9%)	13 (26%)	0.078	39 (78%)	0.342	26 (52%)	0.165
From 2001 to 10,000	27 (23.7%)	6 (22.2%)		18 (66.7%)		9 (33.3%)	
Higher than 10,000	37 (32.4%)	3 (8.1%)	0.078	24 (64.9%)		13 (35.1%)	
Clinical classification	*n* = 116						
HBeAg-positive chronic infection	7 (6%)	0	0.030	6 (85.7%)	0.117	4 (57.1%)	0.639
HBeAg-positive chronic hepatitis	8 (6.9%)	0		3 (37.5%)		3 (37.5%)	
HBeAg-negative chronic hepatitis	41 (35.3%)	6 (14.6%)		30 (73.2%)		15 (36.6%)	
HBeAg-negative chronic infection	60 (51.7%)	16 (26.7%)		44 (73.3%)		28 (46.7%)	
Transmission route	*n* = 116						
Intrafamilial	70 (60.3%)	14 (20%)	0.725	46 (65.7%)	0.081	27 (38.6%)	0.224
Others	46 (39.7%)	8 (17.4%)		37 (80.4%)		23 (50.0%)	

* missing viral load data in two patients.

## Data Availability

The authors agree to provide the study data upon request.

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
