# Peer review of "Core Promoter and Pre-Core Variants of the Hepatitis B Virus (HBV) Are Frequent in Chronic Hepatitis B HBeAg-Negative Patients Infected by Genotypes A and D"

_viruses, 2023, doi:10.3390/v15122339_

Round 1

Reviewer 1 Report

Comments and Suggestions for Authors

In the current manuscript, Reuter T et al showed the BCP/PC variants of subgenotypes A1 and D3 HBV and their association with clinical characteristics in Brazil. The work has some clinical value, but there are some issues should be solved in the manuscript. Please see below for a list of points:

1.      Since the presence of G1764A was associated with A1 subgenotype (p=0.03) and PC variants with D3 (p < 0.01), HBV DNA level and HBeAg status (or HBeAg level) should be compared between subgenotypes A1 and D3 to better understand the effect of G1764A and PC mutations in HBV replication.

2.      The author mentioned core promoter mutations other than those at 1762/1764 appear to upregulate viral DNA replication in Lines 283-286. How about the A1766T, T1768A, C1766T, T1753C or other core promoter mutations in the sample population of this study, and the correlations of these mutations and HBV DNA levels?

3.      In Line 36, the sentence “The classic BCP/PC variants are common in the chronic clinical HBeAg phase” was not clearly expressed.

Comments on the Quality of English Language

    There are many language errors and inconsistencies in the manuscript, such as pre-core/pre core in Line 26/Line 39 and Line 51/Line 62, UI/mL in Line 131, UI in all Tables, HB and Ag status in all Tables, 1764/1764 in Line 207/Line 223.

Author Response

Dear Sir

Thank you very much for the careful review of the present manuscript.

We have deeply reviewed it and carried out a careful review of the English language.

During this review, we concluded that most of the conclusions driven to subgenotypes A1 and D3 could be enlarged to genotypes A and D and decided to emphasize this broader conclusion.

We will respond point to point to your comments:

  1. Since the presence of G1764A was associated with A1 subgenotype (p=0.03) and PC variants with D3 (p < 0.01), HBV DNA level and HBeAg status (or HBeAg level) should be compared between subgenotypes A1 and D3 to better understand the effect of G1764A and PC mutations in HBV replication.

As shown in Table 1, there was not any relation of G1764A with HBVDNA level or HBeAg status. In Table 2, we did show a statistical significance of HBV DNA level and PC mutations.

When we analyze both BCP+PC mutations together, HBV DNA was lower than 2000 UI/mL especially when both BCP/PC mutations were present (p<0.001) or when only PC mutations were detected (p=0.047).

It is a very important issue that was cited in the discussion “Interestingly, among our samples, HBV DNA was lower than 2000 IU/mL especially when both BCP/PC mutations were present (p<0.001) or when only PC mutations were detected (p=0.047), reinforcing their role in the viral replication.” This phrase was also added to the end of the paper abstract. We would like to thank the reviewer for drawing attention to this interesting finding.

  1. The author mentioned core promoter mutations other than those at 1762/1764 appear to upregulate viral DNA replication in Lines 283-286. How about the A1766T, T1768A, C1766T, T1753C or other core promoter mutations in the sample population of this study, and the correlations of these mutations and HBV DNA levels?

These other mutations have been analyzed but we did not find any correlation among them to age, HBeAg status, HBV genotype and subgenotype, HBV-DNA level, clinical classification and transmission route. This phrase was added to our discussion.

  1. In Line 36, the sentence “The classic BCP/PC variants are common in the chronic clinical HBeAg phase” was not clearly expressed.

This phrase was removed after the extensive review the paper was submitted.

  1. There are many language errors and inconsistencies in the manuscript, such as pre-core/pre core in Line 26/Line 39 and Line 51/Line 62, UI/mL in Line 131, UI in all Tables, HB and Ag status in all Tables, 1764/1764 in Line 207/Line 223.

The paper was submitted to an extensive review and these inconsistencies where removed.

Thank you very much for your careful review that increased the quality of the paper.

We hope to have answered all the points raised.

We are attaching a new version of the paper.  

Sincerely yours,

João Renato Rebello Pinho

Reviewer 2 Report

Comments and Suggestions for Authors

Reuter et al. reported that the BCP variant G1764A and the non-classical PC variants G1862T/G1888A, C, or T are associated with subgenotype A1 and inactive HBV carriers. Authors should make corrections of abstract below and ask native English speaker to edit their manuscript before submission.

1.      What does it mean “ Seventy-two percent were brown/black, subgenotype A1/D3 in 73.3%/20.7%.”?

2.      What does it mean “HBeAg 31 (87.4%) population”?

3.      What does it mean “high (69%24%)”?

Comments on the Quality of English Language

Reuter et al. reported that the BCP variant G1764A and the non-classical PC variants G1862T/G1888A, C, or T are associated with subgenotype A1 and inactive HBV carriers. Authors should make corrections of abstract below and ask native English speaker to edit their manuscript before submission.

1.      What does it mean “ Seventy-two percent were brown/black, subgenotype A1/D3 in 73.3%/20.7%.”?

2.      What does it mean “HBeAg 31 (87.4%) population”?

3.      What does it mean “high (69%24%)”?

Author Response

Dear Sir

Thank you very much for the careful review of the present manuscript.

We have deeply reviewed it and carried out a careful review of the English language.

During this review, we concluded that most of the conclusions driven to subgenotypes A1 and D3 could be enlarged to genotypes A and D and decided to emphasize this broader conclusion.

After the extensive review, the points below raised by the reviewer have been solved.

Reuter et al. reported that the BCP variant G1764A and the non-classical PC variants G1862T/G1888A, C, or T are associated with subgenotype A1 and inactive HBV carriers. Authors should make corrections of abstract below and ask native English speaker to edit their manuscript before submission.

  1. What does it mean “ Seventy-two percent were brown/black, subgenotype A1/D3 in 73.3%/20.7%.”?
  2. What does it mean “HBeAg 31 (87.4%) population”?
  3. What does it mean “high (69%24%)”?

The paper was submitted to an extensive review and these inconsistencies where removed.

Thank you very much for your careful review that increased the quality of the paper.

We hope to have answered all the points raised.

We are attaching a new version of the paper.  

Sincerely yours,

João Renato Rebello Pinho

Reviewer 3 Report

Comments and Suggestions for Authors

Dear Authors

I have the following comments

1. What was the treatment exposure status of the participants. Please give details of the antiviral use along with their duration

2. How the intrafamilial transmission was ascertained?

3. Whether the HBsAg positive family members where also examined for genotype and mutations?

4. Please classify the clinical disease according to any of the classification given by either AASLD or EASL but do not mix them, it will confusing for the reader

5. Prior treatment status and antiviral drug exposure status shall also be used as a variable in Tables 1, 2, and 3

6. Table 1: what is HB and Ag status? what does'outros' means?

7. All the tables: International units are written as IU instead of UI. 

Comments on the Quality of English Language

Moderate language correction is needed

Author Response

Dear Sir

Thank you very much for the careful review of the present manuscript.

We have deeply reviewed it and carried out a careful review of the English language.

During this review, we concluded that most of the conclusions driven to subgenotypes A1 and D3 could be enlarged to genotypes A and D and decided to emphasize this broader conclusion.

Dear Authors

I have the following comments

  1. What was the treatment exposure status of the participants. Please give details of the antiviral use along with their duration

The paper was rewritten emphasizing its novelties; the most important results are the characterization of the mutations in precore and basal promoter regions. We did not give more details on the antiviral use along the follow up as this data have been detailed in a previous paper that was cited  in the first paragraph of “materials and methods” “Demographic, epidemiological, clinical, treatment and laboratory characteristics were detailed in a previous publication involving this same population [4].” –

Reuter T.Q., Gomes-Gouvea M., Chuffi S., Duque U.H., Carvalho J.A., Perini W., Queiroz M.M., Segal I.M., Azevedo R.S., Pinho J.R.R. Hepatitis B virus genotypes and subgenotypes and the natural history and epidemiology of hepatitis B. Ann. Hepatol. 2022, Suppl 1, 100574.

  1. How the intrafamilial transmission was ascertained?

intrafamilial transmission was detailed in the paper cited above:

Hepatitis B transmission route was categorized as:  

Mother- to-child-transmission (MTCT) or vertical, defined when the mother was HBsAg positive; 

intrafamilial, when the father was HBsAg positive or had chronic hepatitis, with a susceptible seronegative mother, or serological evidence of a chronic hepatitis B patient among relatives living in the household of the index case;  

parenteral, from the use of injectable or inhaled drugs or by blood transfusion;  

sexual, in individuals with unsafe sex, multiple partners (more than two sexual partners in the 6 months prior to diagnosis), absence of family history and negative HBV infection markers in parents, 

unknown, when the individual did not fill any criteria above or when there was no possibility of family investigation.  

When more than one criterion was found, the one with the greatest potential for infectivity was considered. 

  1. Whether the HBsAg positive family members where also examined for genotype and mutations?

Yes, all the family members were examined for their HBV status, including genotypes and mutation if they were HBV DNA positive

  1. Please classify the clinical disease according to any of the classification given by either AASLD or EASL but do not mix them, it will confusing for the reader

We classify the disease using EASL.

  1. Prior treatment status and antiviral drug exposure status shall also be used as a variable in Tables 1, 2, and 3

We agree that these data are very relevant and might relate to some evolution markers of the patients that should be addressed in another publication.

  1. Table 1: what is HB and Ag status? what does'outros' means?

The paper was submitted to an extensive review and these inconsistencies where removed.

  1. All the tables: International units are written as IU instead of UI.

The paper was submitted to an extensive review and these inconsistencies where removed.

We are attaching a new version of the paper.  

Thank you very much for your careful review that increased the quality of the paper.

We hope to have answered all the points raised.

Sincerely yours,

João Renato Rebello Pinho

Round 2

Reviewer 1 Report

Comments and Suggestions for Authors

None

Comments on the Quality of English Language

None

Author Response

Dear Sir

We have deeply reviewed again the English language and the new changes are highlighted in yellow

Thank you very much for your careful review which increased the quality of the paper.

Sincerely yours,

João Renato Rebello Pinho

Reviewer 2 Report

Comments and Suggestions for Authors

Authors should analyze the full-length sequence of the samples and show the phylogenetic analysis of HBV strain.

Author Response

Dear Sir

We have deeply reviewed again the English language and the new changes are highlighted in yellow.

These changes will make the description of the methodology, the results presentation, and therefore the conclusions clearer to the readers.

The idea to analyze full-length HBV genomic sequences is an interesting idea, but it was not planned in this project.

Phylogenetical analysis of the S/POL sequences was carried out for the whole population and was utilized in the previously published paper (1). “To identify the HBV genotypes and subgenotypes of each case and to study the phylogenetic relationships between the characterized sequences, the alignment was submitted to phylogenetic analysis using the BEAST 1.8.3 program (8) with 10,000,000 steps of the Markov Monte Carlo chain (MCMC) and sampling every 1,000 steps, and rejecting the first 1,000,000 steps as burn-in.”  

  • Reuter T.Q., Gomes-Gouvea M., Chuffi S., Duque U.H., Carvalho J.A., Perini W., Queiroz M.M., Segal I.M., Azevedo R.S., Pinho J.R.R. Hepatitis B virus genotypes and subgenotypes and the natural history and epidemiology of hepatitis B. Hepatol. 2022, Suppl 1, 100574)

Thank you very much for your careful review which increased the quality of the paper.

Sincerely yours,

João Renato Rebello Pinho

Reviewer 3 Report

Comments and Suggestions for Authors

Dear Editor

In my opinion, we need that data on treatment exposure status of the participants. Please give details of the antiviral use along with their duration.

Mutation analysis, in context of HBV, are relevnat only if we know their treatmnet status

Comments on the Quality of English Language

It is ok

Author Response

Dear Sir

We have deeply reviewed again the English language and the new changes are highlighted in yellow.

These changes will make the description of the methodology, the results presentation, and therefore the conclusions clearer to the readers.

As previously stated, we agree that prior treatment exposure might be related to the presence of some of these mutations but this data was not previewed to be utilized in the present publication. General data on previous treatment were shown in a previous publication involving this same population [1]

  1. Reuter T.Q., Gomes-Gouvea M., Chuffi S., Duque U.H., Carvalho J.A., Perini W., Queiroz M.M., Segal I.M., Azevedo R.S., Pinho J.R.R. Hepatitis B virus genotypes and subgenotypes and the natural history and epidemiology of hepatitis B. Hepatol. 2022, Suppl 1, 100574.

Thank you very much for your careful review which increased the quality of the paper.

We hope to have answered all the points raised.

Sincerely yours,

João Renato Rebello Pinho